# Integrative analysis of single-cell RNA-seq and gut microbiome metabarcoding data elucidates macrophage dysfunction in mice with DSS-induced ulcerative colitis

Dawon Hong[1,8], Hyo Keun Kim[2,8], Wonhee Yang [3], Chanjin Yoon[4], Minsoo Kim[5], Chul-Su Yang [6] ✉ & Seokhyun Yoon [7] ✉

Ulcerative colitis (UC) is a significant inflammatory bowel disease caused by an abnormal immune response to gut microbes. However, there are still gaps in our understanding of how immune and metabolic changes specifically contribute to this disease. Our research aims to address this gap by examining mouse colons after inducing ulcerative colitis-like symptoms. Employing single-cell RNA-seq and 16 s rRNA amplicon sequencing to analyze distinct cell clusters and microbiomes in the mouse colon at different time points after induction with dextran sodium sulfate. We observe a significant reduction in epithelial populations during acute colitis, indicating tissue damage, with a partial recovery observed in chronic inflammation. Analyses of cell-cell interactions demonstrate shifts in networking patterns among different cell types during disease progression. Notably, macrophage phenotypes exhibit diversity, with a pronounced polarization towards the pro-inflammatory M1 phenotype in chronic conditions, suggesting the role of macrophage heterogeneity in disease severity. Increased expression of *Nampt* and NOX2 complex subunits in chronic UC macrophages contributes to the inflammatory processes. The chronic UC microbiome exhibits reduced taxonomic diversity compared to healthy conditions and acute UC. The study also highlights the role of T cell differentiation in the context of dysbiosis and its implications in colitis progression, emphasizing the need for targeted interventions to modulate the inflammatory response and immune balance in colitis.

Ulcerative Colitis (UC) is a chronic and debilitating inflammatory disease of the colon and is a distinct condition within a broad group of pathologies termed inflammatory bowel disease (IBD), which includes Crohn's disease (CD) and indeterminate colitis[1,2]. UC is characterized by the loss of intestinal epithelial integrity[3]. Upon tissue damage, the microbiota and their associated products trigger the inflammatory cascade. The dextran sodium sulfate (DSS) colitis model is commonly used in experimental methods to induce UC due to its rapidity, simplicity, reproducibility, and controllability[4–7]. It is possible to establish acute and chronic models of

intestinal inflammation[4]. The cause of intestinal inflammation by modifying the DSS concentration and administration frequency. The exact cause of intestinal inflammation induced by DSS is not yet clear, but it is likely due to damage to the epithelial monolayer that lines the large intestine. This damage allows proinflammatory intestinal contents (e.g., bacteria and their products) to spread into underlying tissue[8]. A comparative analysis of acute colitis (AC) and chronic colitis (CC) is essential to comprehend the shifts in the intestinal environment as the disease progresses. In individuals with IBD, the gut microbiota composition is not adequately induced compared to

[1]RNA Cell Biology Laboratory, Graduate Department of Bioconvergence Engineering, Dankook University, Yongin, Republic of Korea. [2]Dept of Molecular and Life Science and Center for Bionano Intelligence Education and Research, Hanyang University, Ansan-si, Korea. [3]Department of AI-based Convergence, Dankook University, Yongin, Republic of Korea. [4]Dept of Molecular and Life Science and Institute of Natural Science and Technology, Hanyang University, Ansan-si, Korea. [5]Department of Computer Science, College of SW Convergence, Dankook University, Yongin, Republic of Korea. [6]Dept of Medicinal and Life Science and Center for Bionano Intelligence Education and Research, Hanyang University, Ansan-si, Korea. [7]Department of Electronics & Electrical Engineering, College of Engineering, Dankook University, Yongin, Republic of Korea. [8]These authors contributed equally: Dawon Hong, Hyo Keun Kim. ✉e-mail: chulsuyang@hanyang.ac.kr; syoon@dku.edu

healthy condition (HC), often resulting in the transition of acute inflammation into a chronic state[9–11]. Hence, comprehending microbiome alterations as acute inflammation advances into chronic colitis assumes paramount importance.

A growing body of evidence highlights the involvement of the gut microbiota in the pathogenesis and progression of ulcerative colitis. Dysbiotic microbiota in NLRP6 inflammasome-deficient mice has been demonstrated to directly exacerbate colitis severity induced by DSS[12]. Furthermore, dysbiotic communities actively modulate innate immune signaling and the interface between the host and microbiota, thereby affecting disease susceptibility through various immune pathways, ultimately leading to similar host pathology[13,14]. A recent study has even demonstrated the direct activation of NLRP3 inflammasome and IL-1β in LPS-primed murine bone marrow-derived macrophages (BMDMs) by gut bacterial pathogens[15–17]. However, it remains unclear how to change gut dysbiosis in the progression of UC.

On the other hand, single-cell RNA sequencing (scRNA-seq) has emerged as a powerful tool, revolutionizing our understanding of intestinal diseases at the individual cell level[18–20]. This technology allows rapid and precise characterization of gene expression patterns in thousands of cells within the intricate landscape of the intestinal tissue[21–24]. ScRNA-seq illuminates cellular characteristics by analyzing cells with similar phenotypes. It offers insights into growth, development of intestinal organs, clonal cell evolution, and immune cell population alterations. Using public single-cell datasets, our previous study identified key driver genes in macrophages associated with IBD in humans and mouse models through *Nampt*. Notably, we identified and validated that eNAMPT interact with *Tlr4* or *Cybb*/NOX2 and activate NLRP3 inflammasome in IBD tissues[5]. By understanding the relationship between dysbiotic communities and host immune response, we revealed a crucial role for *Nlrp3/Nampt*.

Starting from our previous works, we attempted in this study to dissect the dynamic changes during inflammation using well-prepared scRNA-seq dataset. Specifically, we tried to characterize the pathology and the biological mechanism underlying the ulcerative colitis separately for the acute and chronic colitis. To this end, we prepared mice with DSS-induced acute and chronic colitis to perform single-cell RNA-sequencing for UC colon and 16 s rRNA amplicon sequencing for gut microbiota. Our observations revealed the critical role of *Nlrp3/Nampt* expressed in damaged cells in chronic inflammation. The data we generated will also serve as a foundation for understanding inflammatory diseases.

## Results
### Disruption of the epithelial barrier in DSS-induced colitis
Our experimental workflow encapsulates the systematic approach taken in this study (Supplementary Fig. S1a). Following dextran sodium sulfate (DSS)-mediated induction at distinct time points (days 0, 6, and 33), we conducted a comprehensive single-cell RNA-seq (scRNA-seq) analysis on mouse colon samples. To increase robustness and mitigate interindividual variability, we prepared biological triplicates for days 0 and 6, and quadruplicates for 33 days (Supplementary Fig. S1a). This design allows for a more accurate representation of molecular changes during inflammation progression. Mice were subjected DSS–mediated colitis, which is colitis model that mimics human IBD. The symptoms of colitis, such as weight loss, rectal bleeding, and edema[4], were observed in DSS-induced mice compared with control mice (Fig. 1a, c). Colitis scores were determined based on clinical parameters such as weight loss, stool consistency, and bleeding (Supplementary Table S1 in the Supplementary Information). Different patterns of body weight loss occur in acute and chronic ulcerative colitis (UC). In the case of acute UC, body weight gradually decreases over time. While, in chronic UC, body weight decreases for 12 days and then recovers for 6 days, repeating this cycle three times. (Fig. 1a, c). In macroscopic histological observations, both groups exhibited larger areas of ulceration and crypt loss in the colon, along with disruption of epithelial cells in the acute and chronic colon compared to the control colon. (Fig. 1b, d).

To analyze the molecular intricacies, we used the 10X genomics' chromium method to profile 84,612 cells from the colon of 10 mice at three different time points. Through integrated analysis of the collected data, we identified and characterized 7 distinct cell clusters based on their unique gene expression profiles (Fig. 1e and Supplementary Fig. S1b). To visually depict the alterations in cell composition during inflammation, we quantified the proportion of major cell types at each time point (Fig. 1f). The analysis showed significant shifts, particularly marked by a dramatic reduction in epithelial populations, presumably reflecting tissue damage (Fig. 1f). On the other hand, there was a notable increase in the number of myeloid-cell, T-cell, and B-cell clusters during the inflammation phase, and it did not return to a steady state. Although the difference in the proportion of epithelial cells were not statistically significant due to the limited number of samples, the epithelial barrier status could also be measured in terms of gene and protein level expressions of some related genes since the loss of epithelial barrier integrity in UC is associated with impairment of the junction proteins and increased epithelial apoptosis[25]. After inducing acute inflammation for 6 days, we observed a significant decrease in genes related to epithelial junctions such as *Ocln*, *Cldns*, and *Tjap1*, as well as changes in their expression patterns (Figs. 1g and S1d). Remarkably, in the 33-day chronic inflammation induction model, we noticed a partial reversal of the disruption to the epithelial barrier observed in AC (Fig. 1f, g). This slight restoration of intestinal epithelial cell function emphasizes the potential for recovery in prolonged and repetitive chronic inflammation scenarios. We confirmed that the protein levels of intestinal barrier status biomarkers, including Zonula-1, Claudin-1 (*Cldn*), and Occludin (*Ocln*) shows the disruption of epithelial barrier in both AC and CC mice, while it was slightly restored in CC when compared to AC mice (Fig. 1h). This finding highlights the severity of epithelial barrier disruption in AC compared to the CC model. The results strongly suggest an association between AC and a more pronounced loss or dysfunction of epithelial cell junctions with CC displaying a potential for restoration of epithelial cell functionality.

### Alteration of cell-cell interactions during the progression of UC
Cells constantly interact with each other to orchestrate organismal development, homeostasis, and single-cell function. When cells do not interact properly or misinterpret molecular signals, it can result in the development of diseases. We hypothesized that a dynamic shift in intercellular interactions causes dysregulation and progression of UC. ScRNA-seq is aiding to advance our understanding of UC by comprehensively mapping the cell types and states within a tissue, disentangling changes in the expression of gene and connecting them through cell-cell interactions[26] We mapped receptor-ligand pairs onto cell minor types using CellphoneDB software tool to construct a putative cell-cell interaction network across disease states[27] (Supplementary Data S1). In HC, a robust interconnection was observed between epithelial cells and fibroblasts primarily facilitated by collagen and integrins (Fig. 2a, b). Proper interaction between epithelial and immune cells was maintained and intercellular interaction among immune cells was also appropriately maintained (Fig. 2b). On the other hand, the inferred cell-cell interactions between immune cells exhibited a significant increase in UC compared to healthy conditions (Fig. 2a). Specifically, during AC, the interplay between fibroblasts and epithelial cells was notably decreased (Fig. 2a), whereas it was seen that there was an increased interactions among immune cells, especially with macrophage (Fig. 2a middle). Our analysis also unveiled a noteworthy interaction pattern during AC, specifically highlighting the interplay between chemokines from macrophages and chemokine receptors in B cell, T cell, and fibroblast (Fig. 2c). In the context of CC, there was a notable enrichment of interactions between macrophages and epithelial cells when compared to AC (Fig. 2a right). Furthermore, we observed a restoration of the interactions between epithelial cells and fibroblasts during CC, where ligand-receptor pairs in epithelial cells and fibroblasts were resembling those of HC (Fig. 2d). Different from interactions between epithelial cells and fibroblasts, there is a newly facilitated interaction between epithelial cell and macrophages, mediated by chemokines and chemokine receptors, including *Nampt* and *Cybb*

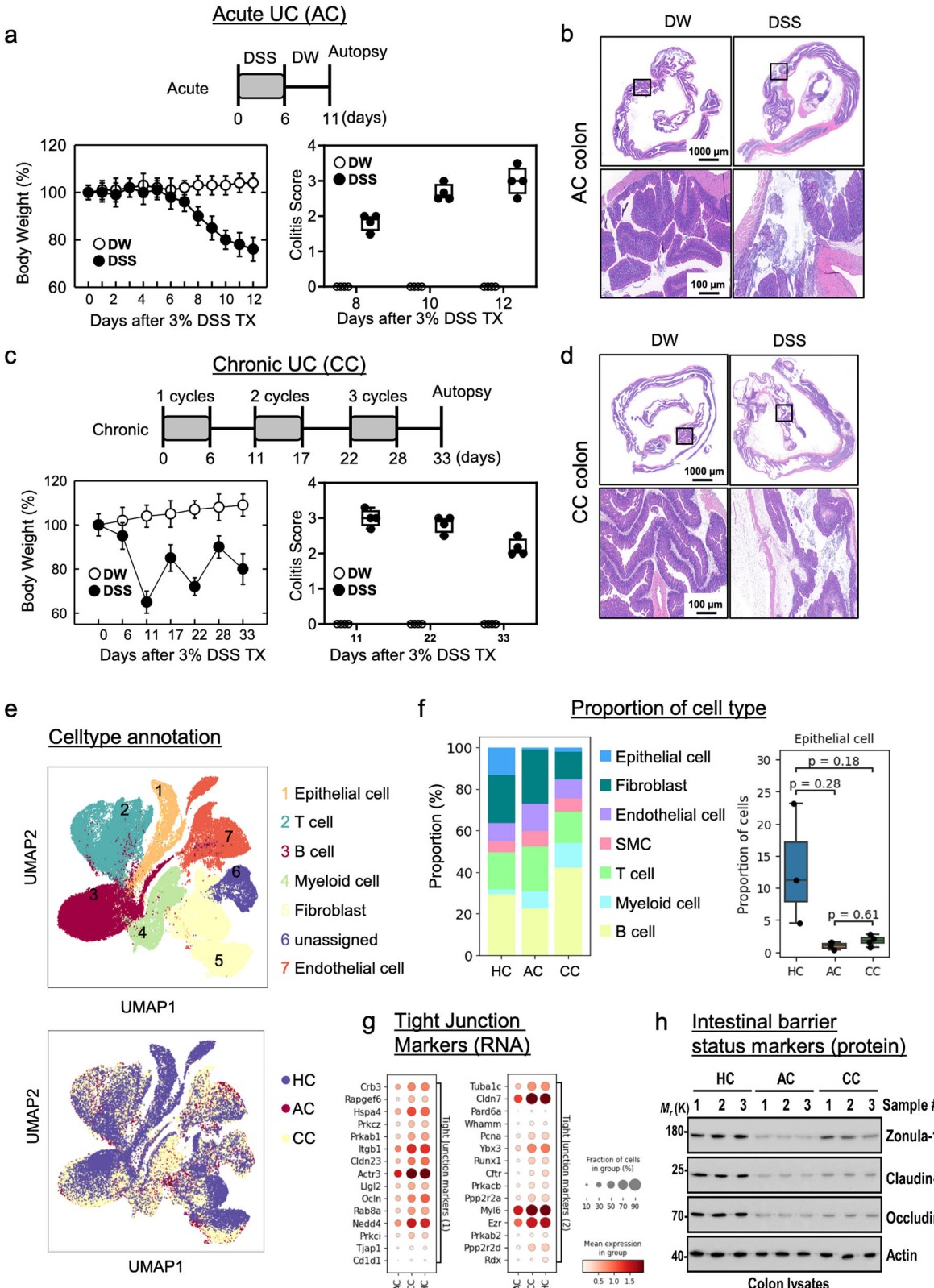

(also known as gp91phox) receptors (Fig. 2d). *Nampt*, also known as B-cell pre-colony enhancer or visfatin, has cytokine-like functions that enhance B-cell pre-colonies, monocyte growth, and macrophage survival[28]. These findings suggest a dynamic shift in interaction patterns among these cell types as colitis progresses. Therefore, UC progress is switched by intercellular interactions. In AC, various immune cells, including macrophages, pass through the severely damaged epithelial barrier, which results in intercellular interaction. In CC, interactions were enforced not only between macrophages and epithelial cells but also between macrophages and fibroblasts.

**Fig. 1 | Single-cell RNA-seq profiling of epithelial cell in DSS-induced mouse colitis. a** Mice were treated with 3% DSS for 6 days and evaluated at day 11 for acute colitis. The relative body weight of mice treated with 3% DSS and DW. Colitis scores were determined based on clinical parameters such as weight loss, stool consistency, and bleeding. **b** Histological appearance of the colon at day 11 in mice with 3% DSS compared with DW. **c** Scheme of the chronic colitis (CC) model induced by 3% DSS. Relative body weight changes during the experiment were measured. Colitis scores were determined based on clinical parameters such as weight loss, stool consistency, and bleeding. **d** Histological appearance of the colon at the endpoint of the experiment in mice with 3% DSS compared with DW. **e** UMAP plots displaying cell populations from all participant mice colored by major cell type (upper plot) and by condition (bottom panel; HC indicates healthy conditions, AC acute colitis, and CC chronic colitis. **f** Bar plots showing proportion of major cell type in each condition. **g** Dot plot depicting the expression profiles of cell barrier-related genes in epithelial cell population. **h** Western blotting results showing the protein level expression of intestinal barrier status markers. DW distilled water, DSS dextran sodium sulfate. The uncropped/unedited blot images were added in the Supplementary Information (Supplementary Fig. S5).

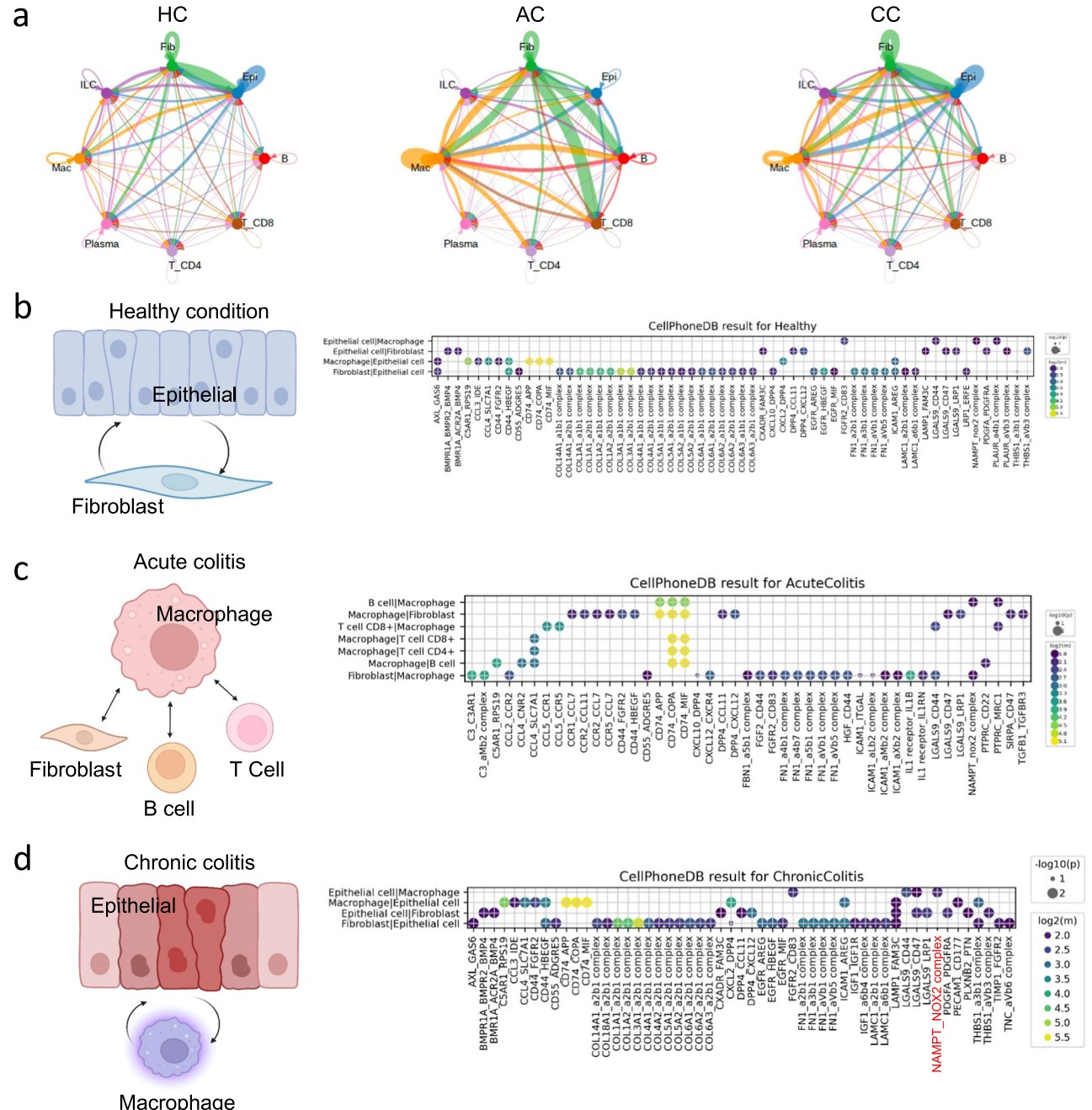

**Fig. 2 | Inferred intercellular interactions in DSS-induced UC. a** Circle plots showing the number of interactions between different cell types. Linewidth corresponds to the number of interactions while line color is the source cell type.

**b–d** Ligand-receptor pairs detected with CellPhoneDB are displayed in a bubble plot for HC (B), AC (C) and CC (D). In **b–d**, the illustrations on the left were created with BioRender.com.

## ScRNA-seq reveals different populations of inflammatory macrophage through the progression of DSS-induced UC

Numerous studies emphasize the pivotal role of macrophages differentiation into diverse phenotypes in modulating the immune response and inflammation in various diseases, including ulcerative colitis[29,30]. The tissue microenvironment plays a crucial role in directing the polarization of monocytes towards either pro-inflammatory (M1) or pro-resolving (M2) macrophage phenotype[31]. Furthermore, evidences suggest a causal association between impaired resolution of intestinal inflammation and altered monocyte-macrophage differentiation[32]. In this study, we investigated the differentiation patterns of macrophages, tracking the progression of UC. Notable alterations in macrophage phenotypes within the colonic environment were observed (Fig. 3 and Supplementary Fig. S2). In AC, macrophages exhibited differentiation into M1 and M2B phenotype (Fig. 3b). Although the population differences were not statistically significant due to the limited number of samples, the tendency shows the increased population of M2B in AC especially when compared to CC. (Fig. 3b). Different from AC, on the other hand, CC was characterized by a pronounced polarization towards the pro-inflammatory macrophage M1 phenotype, suggesting increased inflammation. These results highlight the enhanced macrophage heterogeneity in DSS-induced UC, particularly prominent in CC. Macrophages, known for promoting inflammation, were found to significantly increase *Il1b*, *Cxcl16*, *Ccl19*, *Il18*, and *Ccl6* chemokines and cytokines in CC, thereby contributing to the inflammatory milieu. Interestingly, AC also showed increases in these chemokines and cytokines, albeit to a lesser extent than M1 and M2B in CC (Fig. 3c). Therefore, even though macrophages exhibit less polarized towards M1 in AC, the drastic increase in inflammatory cytokines such as *Il1b* and *Tgfb1* contributes to AC. *Il1b* which is a key mediator released by macrophages in inflammation and immunological responses, plays a crucial role in promoting inflammation in patients with IBD[33–35]. According to our scRNA-seq analysis, we have confirmed that *Il1b* increased in M1, M2A, M2B, and M2C in both AC and CC when compared to HC (Fig. 3d). Our validation experiments also showed a significant increase of IL-1β protein in both macrophages and colon lysates (Fig. 3e). We also conducted a Gene Ontology analysis based on the differentially expressed genes (DEG) in macrophage (Fig. 3f). The results showed increased activity in pathways related to inflammation, specifically NF-κB and NOD-like receptor signaling pathways, indicating an increase in inflammation-associated targets in UC samples. In addition, it was observed that pathways associated with colorectal and pancreatic cancer were enhanced in CC (Supplementary Fig. S2c). It might imply the potential progression of CC to cancer and is consistent with recent reports that long-term, self-perpetuating intestinal inflammation may result in colorectal cancer (CRC)[36]. Taken together, it underscores particularly the polarization towards pro-inflammatory M1 phenotype in CC, indicating macrophage heterogeneity in disease severity.

## The overexpression of eNAMPT and NOX2 subunits leads to the progression of severe colitis induced by DSS

A recent study revealed "nicotinate and nicotinamide metabolism" as the top regulated pathway distinguishing inflamed tissues. NAD+ metabolism maintains intestinal homeostasis[37]. During inflammation, serum NAD+ levels increased three-fold in IBD patients compared to healthy individuals[37]. NAMPT, a key enzyme in the NAD+ salvage pathway, is upregulated in IBD[5,28,38]. Blocking this enzyme has shown promise in alleviating experimental colitis, highlighting its potential as a therapeutic target[39,40]. Our previous data revealed that extracellular NAMPT (eNAMPT) directly interacts with *Cybb* and *Tlr4* in activated NLRP3 inflammasomes, suggesting a potential mechanism for its involvement in inflammation[5] (Supplementary Fig. S3a). Building on our previous findings, we explored *Nampt* expression and subunit of NOX2 complex in a subtype of immune cells (Fig. 4a). *Nampt* expression exhibited a partial increase in UC, even though the *p*-value did not reach significance (Fig. 4b). Surprisingly, the subunits of NOX2 complex, including *Cybb* (also known as gp91phox), *Cyba* (p22phox), *Ncf1* (p47phox), *Ncf2* (p67phox), and

*Ncf4*(p40phox), showed a significant enhancement in mRNA expression level only in CC, but not in AC (Fig. 4b and Supplementary Fig. S3b). Furthermore, Western blot analyses confirmed the protein expression of *Nampt*, *Cybb*, *Ncf1*, *Ncf2*, and *Ncf4* in macrophages and colon tissue lysates, which were elevated in both AC and CC when compared to HC (Fig. 4c, d). Given the increased levels of *Nampt*, further investigations were conducted to determine the protein level expression of eNAMPT in macrophages, revealing an elevation (Fig. 4d). This resulted in an increase of NAD+ in UC due to the increase in *Nampt* and decrease in *Sirt1* which is an enzyme involved in NAD+ consumption[41,42] (Supplementary Fig. S3c, d). These findings suggest that the increased expression of *Nampt* and NOX2 complex subunits, particularly in the CC, may contribute significantly to the inflammatory processes associated with the progression of UC.

Previous studies have shown that NOX2-mediated generation of reactive oxygen species (ROS) activate NLRP3 inflammasomes, leading to IL-1β activation[5]. In our study, we tested *Nlrp3* expression levels during the progression of colitis. We confirmed that the priming step initiator, including *Tlr4* and *Nfkb1*, and NLRP3 inflammasome components including of *Nlrp3*, *Casp1* and *Nek7* were increased in CC, but not in AC (Fig. 4e and S3e). We identified over-expression of ROS-scavenging and hypoxia-alleviating genes, including *Sod2*, *Neat1*, and *Hif1a*, but not *Sirt1* (Supplementary Fig. S3c, d). Moreover, the levels of IL-1β, which is activated by the NLRP3 inflammasome, were significantly elevated in both acute and chronic UC (Figs. 3e and 4e). Taken together, the upregulation of eNAMPT and NOX2 subunits led to NLRP3 inflammasome activation in colitis macrophages, which contributes to severe colitis.

## The intestinal microbiome undergoes changes as colitis progresses

Recent studies have shown that the gut microbiota functions as a metabolic organ, actively contributing to human health by participating in various physiological processes within the host[43,44]. Consequently, the composition of the gut microbial communities differs significantly between healthy individuals and IBD patients, a difference commonly referred to as "gut dysbiosis" meaning imbalance of gut microbiota[45]. Understanding these compositional changes in the gut microbiota is crucial for developing effective strategies for diagnosis and treatment of the disease. We hypothesized that gut dysbiosis might trigger an overexpression of *Nampt* in the host environment, accelerating inflammation. To analyze the microbial composition of these samples, we performed 16 S rRNA gene amplicon sequencing of the healthy, and DSS-induced UC. Multi-dimensional scaling (MDS) plot for the composition of the microbiotas showed well-clustered samples of the healthy, acute, and chronic UC (Fig. 5a). The gut microbiomes in CC exhibited a general reduction in taxonomic diversity compared to those in HC, even though AC did not show significant changes, consistent with the broader trend of decreased species diversity typically observed in IBD[9,11] (Fig. 5b). At the phylum level, the intestinal microbiome indicated a gradual decrease of *Firmicutes* and a significant increase of *Proteobacteria* in UC (Fig. 5c). Comparing metagenome between AC and HC (Fig. 5d upper) and between CC and HC (Fig. 5d lower), although the metagenomes of AC showed a large overlap with HC microbiota (represented by the orange color), indicating only a little difference in diversity, there were variations in the type of microbiota belonging to the family (Fig. 5d upper). Conversely, the coverage range of the metagenome of CC was very narrow and had quite small overlap with that of HC, indicating a reduced microbiota diversity (Fig. 5d lower). Healthy individuals had a majority of *Lactobacillaceae* (~54.6%) in their microbiota, a family within the phylum *Firmicutes* (Fig. 5e, f). The *Lactobacillus* genus is a prominent member of this family and known to alleviate damage to the intestinal epithelial cell barrier and have protective effects in ulcerative colitis[46,47]. The AC metagenome exhibited a significant decrease in *Lactobacillaceae* while a marked increase in *Turicibacteraceae* (Fig. 5f) indicating that imbalance of *Lactobacillaceae* and *Turicibacteraceae* may predispose individuals to acute colitis. On the other hand, in CC, the abundance of *Bacteroidaceae* and *Enterobacteriaceae* increased (39.3% and 10.7% respectively) (Fig. 5e, f).

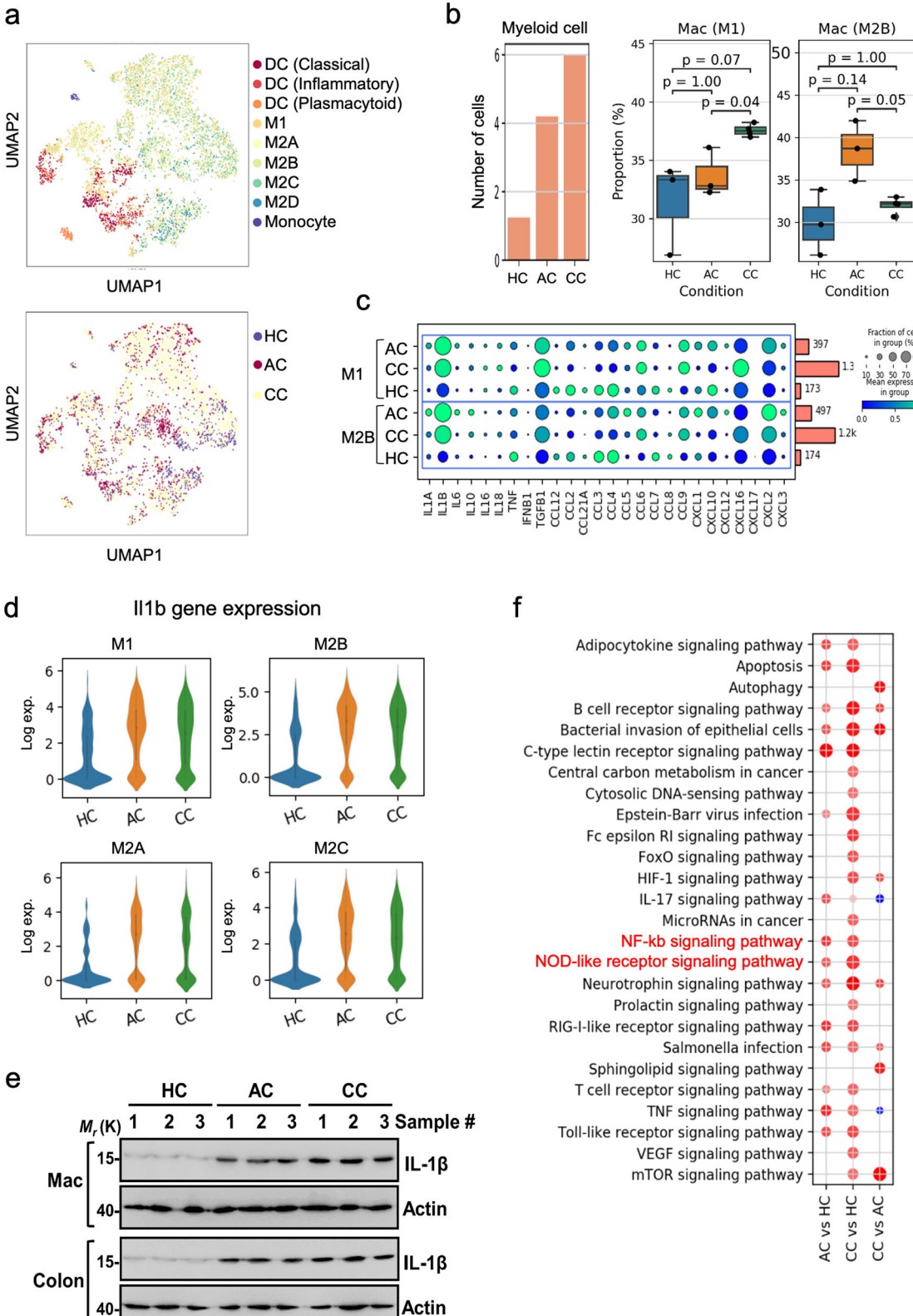

**Fig. 3 | Analysis of myeloid cell subsets in healthy and DSS-induced UC. a** UMAP plot of scRNA-seq data for myeloid cell clusters in HC (*n* = 3), AC (*n* = 3), and CC (*n* = 4). **b** Bar plot showing cell type enrichment analysis. **c** Dot blot showing expression of DEGs for M1 macrophages in HC, AC, and CC. **d** Violin plots of *Il1b* expression in heterogenous polarized macrophages in AC, CC, and HC. **e** Western blotting result showing the IL-1β protein level expression in macrophages and colon tissue lysates. **f** Functional annotation enrichment analyses of upregulated genes in AC and CC. The uncropped/unedited blot images were added in the Supplementary Information (Supplementary Fig. S5).

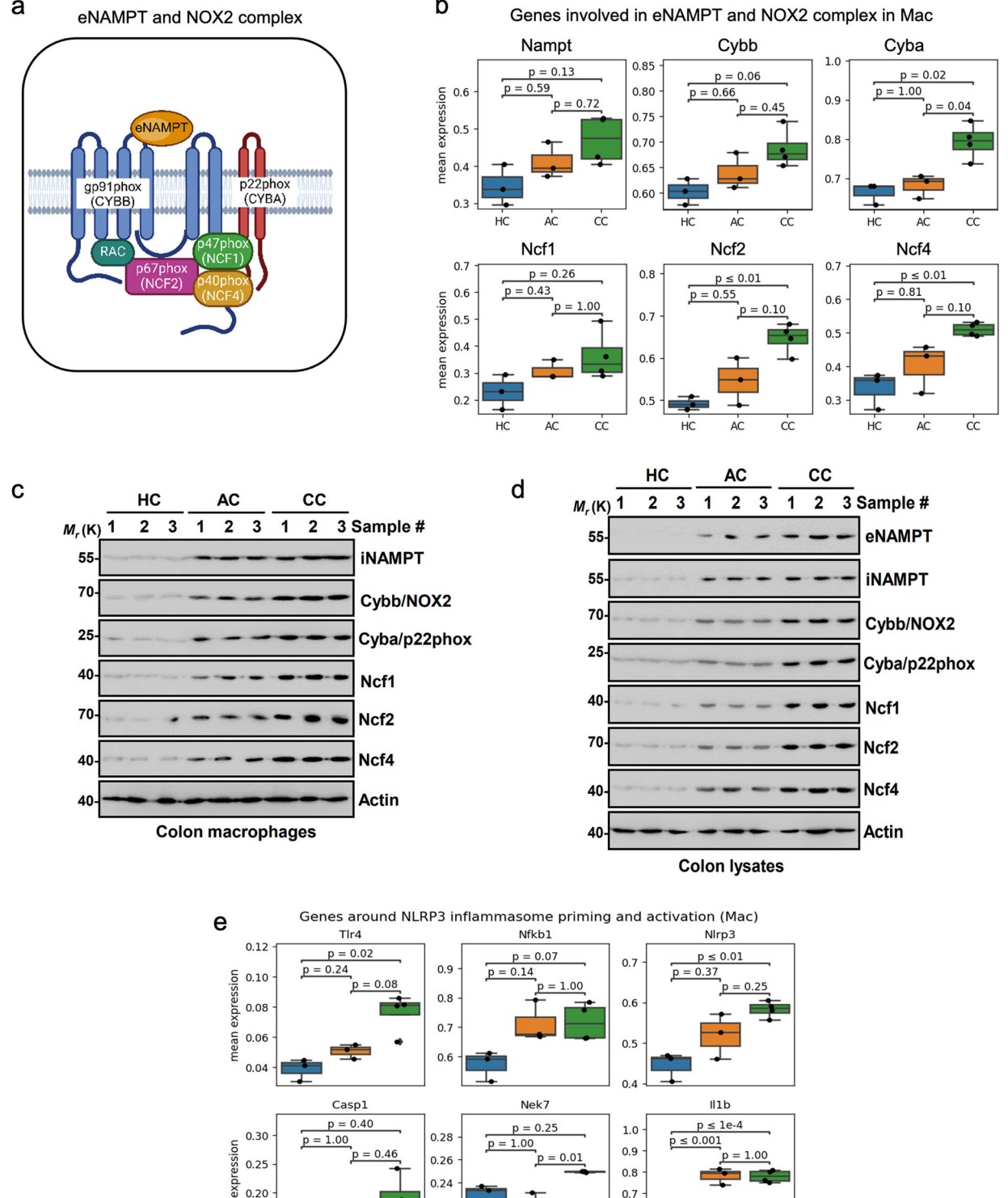

**Fig. 4 | Enhancement of Nampt and NOX2-complex association led to IL-1β levels in both AC and CC macrophages. a** Schematic diagram of eNAMPT and NOX2 complex interaction. The illustrations were created with BioRender.com. **b** Box plot showing the expression profile of genes encoding *Nampt* and NOX2 subunits in HC, AC, and CC macrophages. **c, d** Western blotting results indicating the upregulation of *Nampt*, *Cybb*, Cyba, *Ncf1*, *Ncf2*, and *Ncf4* in colon macrophages and colon lysates. **e** Box plots depicting the expression profiles of NLRP3 inflammasome priming and activation genes in macrophages, as determined by scRNA-seq in AC, CC, and HC. The uncropped/unedited blot images were added in the Supplementary Information (Supplementary Fig. S5).

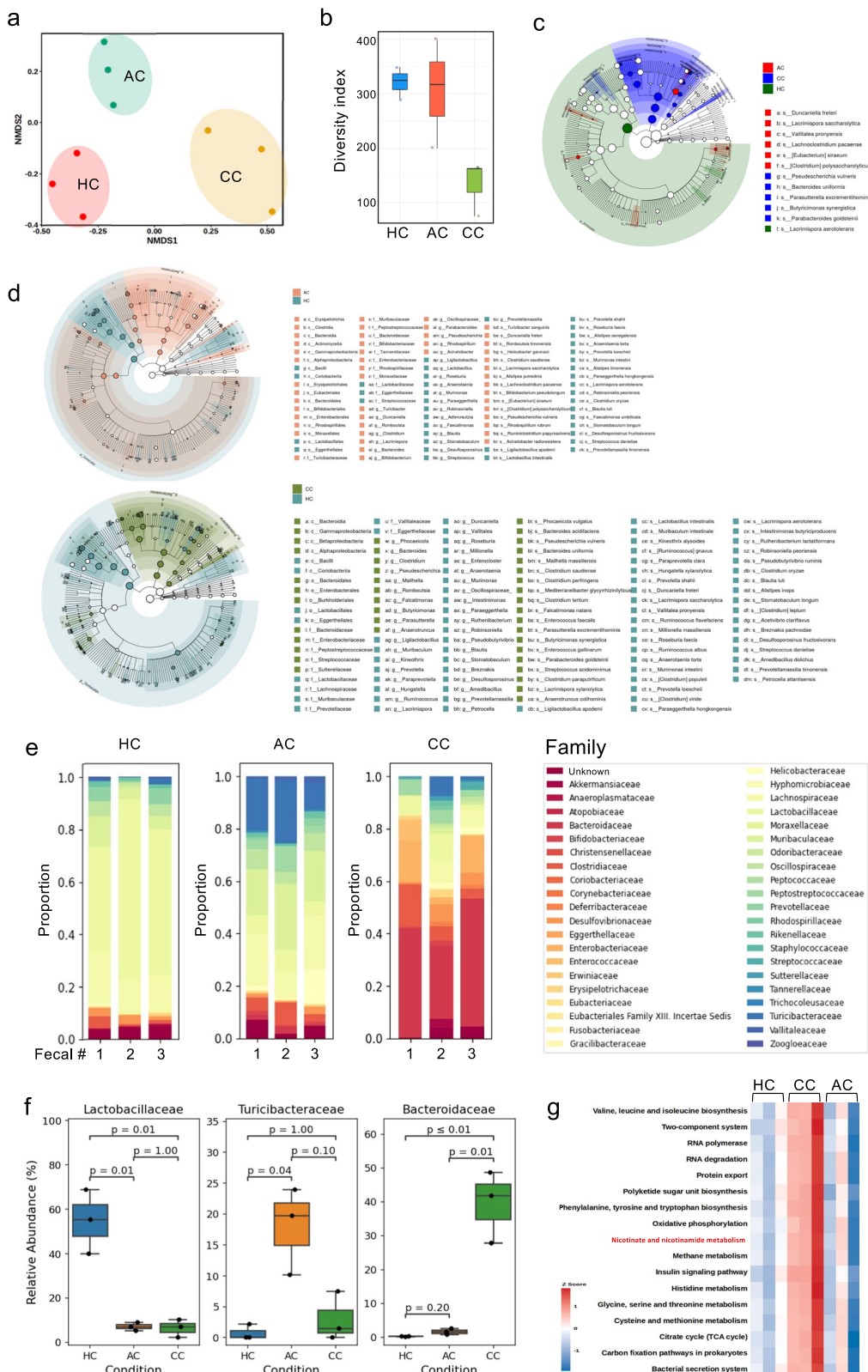

**Fig. 5 | Analysis of commensal microbiota from the progression of DSS-induced colitis. a** Multi-dimensional Scaling (MDS) plots showing clustering of microbiome profiles based on 16s rRNA sequencing. **b** Box plot showing the diversity index. **c, d** Cladogram showing different abundant taxa between samples from different groups (**c**). The upper panel of **d** illustrates AC versus HC, while the bottom panel of **d** shows CC versus HC. The node size represents the difference in relative abundance. p, phylum; c class, o order, f family, g genus. **e** Bar plot showing the relative abundance and distribution of the family in HC, DSS-induced AC (11 days), and DSS-induced CC (33 days). **f** Bar plot showing the relative abundance of three microbiota families, Lactobacillaceae, Turicibacteraceae, and Bacteroidaceae, in HC, AC, and CC. **g** Heatmap of top17 KEGG pathways for HC, AC, and CC based on the normalized enrichment score (NES).

Numerous proposed roles of *Bacteroidaceae* in colitis pathogenesis have been identified. For example, animal studies demonstrate the crucial role of *Bacteroides* in exacerbating colitis symptoms[48–50]. Also, extensive research highlights the significant contribution of *Enterobacteriaceae* to IBD pathogenesis[51] and it seems that alterations in the abundance of *Bacteriodaceae* and *Enterobacteriaceae* may accelerate chronic colitis, resulting in sustained dysbiosis and an altered pathogenic microbiota ratio and eventually leading to colon cancer.

These alterations in intestinal microbiome composition in acute and chronic UC can have implications for the inflammatory state of the gut. Microbiome communities produces specific metabolites influencing host food digestion, xenobiotic metabolism, and production of variety of bioactive molecules[45,52]. Dysbiosis can alter the pools of available metabolites thereby modifying host-generated signaling molecules[9]. We analyzed the related function of the microbiome in healthy controls and UC using PICRUSt2 software[53]. Intriguingly, dysbiosis upon CC induces many metabolisms pathway, including the nicotinate and nicotinamide metabolism pathway (Fig. 5g). Nicotinate and nicotinamide are precursors for synthesis of nicotinamide adenine dinucleotide (NAD+).There is a growing recognition of the complex relationship between inflammatory diseases, NAD+ metabolism, and intestinal homeostasis[37] and some studies showed association of NAD+ metabolism with inflammatory diseases and IBD[54]. With these previous works, we suggest that activation of nicotinate and nicotinamide metabolism by dysbiosis in CC potentially increase nicotinamide concentration in the gut lamina propria (Fig. 5g). The precursors are also directed to the NAD+ biosynthetic pathway through a salvage pathway that is mediated by *Nampt*. These results imply that high level of *Nampt* in macrophage converts increased nicotinamide (NAM) into nicotinamide mononucleotide (NMN). NMN can then be transported into macrophages, where it can be further converted to NAD+ through intracellular *Nampt*. This increase in excessive NAD+ levels within host immune system can contribute to chronic inflammation.

## The differentiation of T cells in relation to dysbiosis with the microbiome is linked to the progression of colitis

Recent research has shown that the gut microbiome plays a significant role in modulating the immune system, including T cell differentiation, and can have important implications for colitis and other inflammatory bowel diseases (IBD)[55]. The gut microbiome interacts with the immune system in the intestines, which can affect the development and function of T cells[55]. Our scRNA-seq data revealed T cells with distinct expression patterns of subset markers among AC, CC, and HC (Supplementary Figs. S4a, b). The UMAP plots illustrated the annotation of T lymphocyte clusters into subsets based on the expression of canonical subset markers (Figs. 6a and S4a, b). Notably, we observed differentially polarized CD4 + T cells in CC of mouse model (Fig. 6b). That is, regulatory T cells (Treg), T helper 17 (Th17), and follicular helper T cell (Tfh) tend to be enriched in CC, even though the differences from HC were not significant due to the limited number of samples. Comparing the proportion of CD4 + T cell subsets of the UC mouse (Fig. 6b) with those of the human UC patients (Fig. 6c), similar tendencies could be observed in the inflamed samples from the human UC, where they were significant. Note that AC did not exhibit significant differences from HC (Fig. 6b), suggesting that a 6-day treatment with DSS has a partial impact on the microbiome, resulting in minimal or no changes in the polarization of T lymphocytes in AC. Importantly, Tregs are generally recognized for broadly immune suppressive and anti-inflammatory properties[56–58]. However, our data demonstrated an enrichment of the Treg in both mouse and human, consistent with previously published reports[59–61]. This led us to hypothesize that the simultaneous increase in Treg and Th17 cells may contribute to chronic colitis severity. Furthermore, we reasoned that the cytokines secreted by both Treg and Th17 cells play a pivotal role in UC, and an imbalance between these two cell types can accelerate the progression of chronic UC (Fig. 6d, e). Our results showed that certain inflammatory cytokines, including *Ccr2*, *Ccr5*, and *Cxcr5*, are produced by Tregs (Fig. 6d), implying that the impaired suppressive

function of Tregs, as influenced by these cytokines, could lead to uncontrolled activation of proinflammatory cells[60]. Note also that an excessive *Il17* production in Th17 was observed in both CC mouse and inflamed tissues of human UC patients (Fig. 6d, e). All these results suggest severe dysbiosis can lead to an altered composition of Treg and Th17 in CC.

## Discussion

In this work, single-cell RNA-seq and 16 s rRNA amplicon sequencing were performed to investigate the complex dysregulation of the host immune system and gut microbiota during the progression of UC. We attempted to utilize various software tools for scRNA-seq data analysis in order to gain comprehensive understanding of the hidden biological meaning in UC. Our integrated single-cell transcriptomics and gut microbiota metagenomic analyses gave us several new insights into the immunometabolism of UC. First, we observed a significant disruption of the epithelial barrier during acute UC, with partial recovery noted in cases of CC. This disruption of the epithelial barrier resulted in a marked infiltration of innate and adaptive immune cells and microbiota. In AC, the severe epithelial barrier disruption induced by DSS led to the infiltration of macrophages and CD8 + T cells into the lamina propria. Enhanced interaction (increased number of interactions) between epithelial cells and fibroblasts could also be observed as UC progresses. In CC, on the other hand, there was an enhancement in interaction between macrophages and epithelial cells through multiple ligand and receptor interactions, including that of eNAMPT and NOX2 complex. Significant increases in the myeloid cluster were observed with distinct polarization into dendritic cells in AC and shifting toward M1 and M2B macrophages in CC. To gain insight into the functional consequences of UC-associated dysbiosis, microbiome profiles were also compared for each pair of conditions. Interestingly, we identified microbiome diversity significantly decreased in CC, even though it was not the case in AC. We analyzed PICRUSt2 to infer functional profiles from 16 S rRNA amplicon sequencing data. However, since PICRUSt2 analysis is based on taxonomic information, it may be inaccurate. Despite the disrupted diversity and dysbiosis in CC, the production of metabolites through metabolic pathways increased. In UC, activation of the nicotinate and nicotinamide metabolism pathway may induce severe inflammation by cell-cell interactions within the host immune system, such as eNAMPT-NOX2 complex interaction in macrophages. Regarding T cells, we saw that simultaneous increase of Tregs and Th17 or their imbalance may lead to the persistence and deterioration of the disease, resulting from the secretion of certain inflammatory cytokines, such as *Il17*, *Ccr2*, *Ccr5*, and *Cxcr5*[62,63].

These findings indicate a notable shift in the dynamics of cell-cell interactions during UC progression. In a healthy intestinal environment, the epithelial cell barriers demonstrate robustness, facilitating the maintenance of homeostasis between the immune system and gut microbiota. However, following a 6-day DSS treatment, we observed the disruption of the epithelial barriers, mucosal alterations, and infiltration of microbiota into the lamina propria, which triggered the activation of macrophages and T cells in response to pathogenic signals, resulting in a substantial increase in interactions between epithelial cells and the immune cell populations during AC. In the context of CC, we observed a shift in cell-cell interactions between epithelial cells and macrophages, characterized by a heightened interplay among immune cells. The enhanced interaction was mediated through ligand-receptors interactions such as eNAMPT-NOX2 complex.

The role of eNAMPT in promoting inflammation and T cell differentiation in colitis has attracted attention as a potential therapeutic target. Inhibiting eNAMPT activity may represent a strategy to modulate the inflammatory response and T cell differentiation in colitis, potentially providing a new avenue for treatment. It has been shown that eNAMPT can affect T cell differentiation and function[64], specifically promoting the differentiation of naïve T cells into Th17, a subtype of T helper cells that are involved in the production of pro-inflammatory cytokines, particularly IL-17. Th17 has been implicated in the pathogenesis of colitis and other

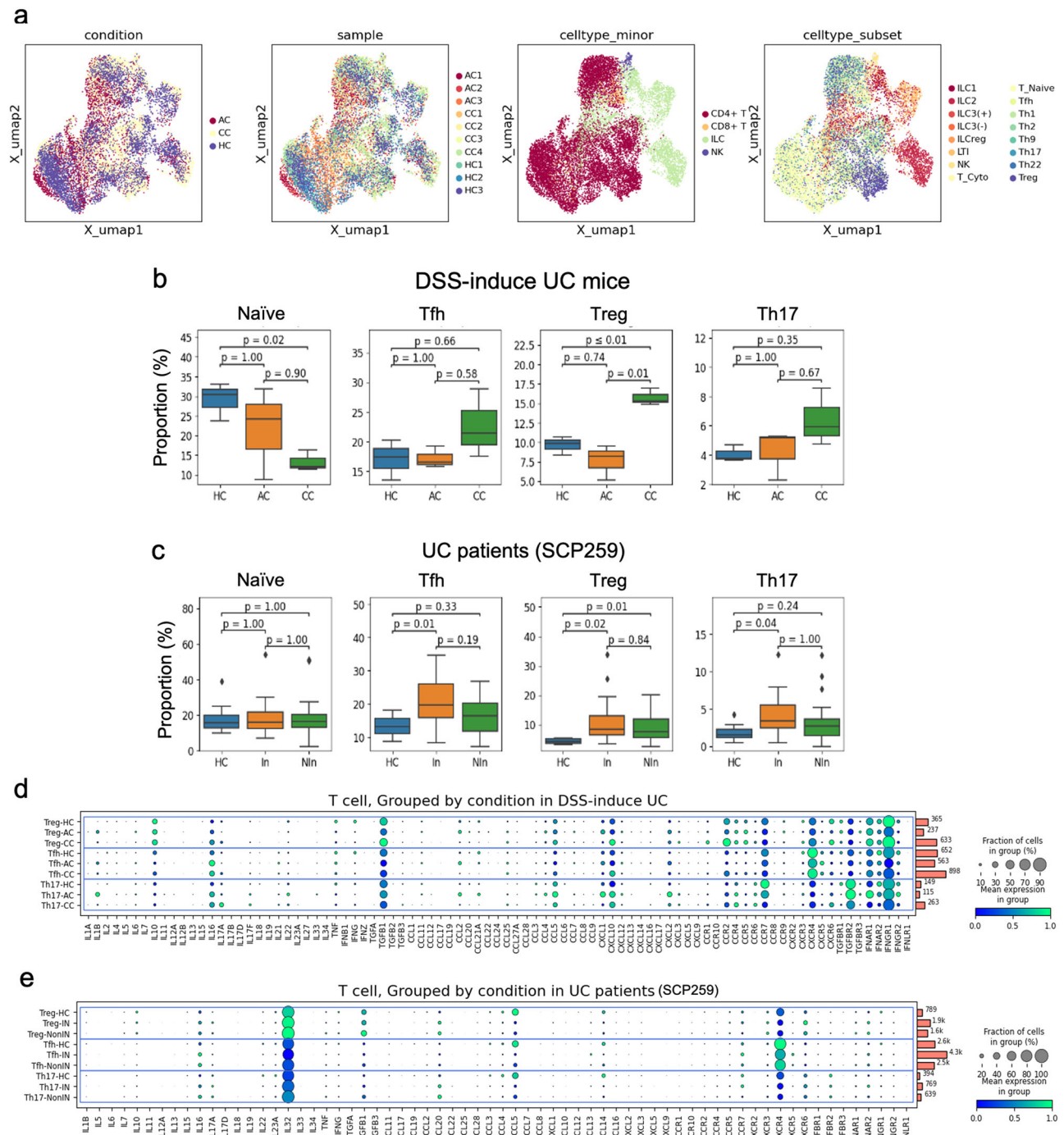

**Fig. 6 | Analysis of T cell subsets in healthy and DSS-induced UC. a** UMAP plot of scRNA-seq data for T cell clusters in HC (*n* = 3), AC (*n* = 3), and CC (*n* = 4). **b, c** Box plots showing cell-type enrichment analysis in DSS-induced UC mouse and human UC patients in SCP259 datasets, respectively. **d, e** Dot blot showing gene expression for Treg, Th17, and Tfh in DSS-induced UC mouse and UC patients in SCP259 datasets, respectively.

autoimmune diseases. Furthermore, we analyzed that the imbalance between Tregs and effector T cells, including Th17 cells, can contribute to the chronic inflammation observed in IBD.

In our metagenome analysis, we identified diversity significantly decreased in CC, but not in AC. Despite no significant change in diversity, specific microbiota such as *Turicibacteraceae* increased exclusively in AC. Since *Turicibacteraceae* was found to have a negative correlation with IL-1β and IL-6 in patients with depressive disorder[65], further validation is required to assess the impact of *Turicibacteraceae* colonization in AC. Another thing to note is that *Sutterella* affects the therapeutic outcomes in UC patients[66], that is, the lower its presence, the better the results we get. Our results

showed a positive association of a *Sutterellaceae* family, *Parasutterella excrementihominis* species, with CC that may cause UC-induced colon cancer[66,67].

Investigating T cells in UC mice and human UC patients, we have found that dysbiosis in CC can affect T cell heterogeneity, which could explain the imbalance of Treg and Th17 observed in human UC patients. This finding is significant as it sheds light on a possible reason behind the T cell imbalance due to dysbiosis. This imbalance can contribute to chronic inflammation in the gut and worsen colitis symptoms. Th17 is involved in the defense against certain pathogens but have also been implicated in the pathogenesis of autoimmune and inflammatory diseases, including colitis.

The increased extent of both Treg and Th17 may correlate with disease severity in IBD since the high levels of Th17 and Th17-associated cytokines are often associated with more severe inflammation and clinical symptoms[68].

In conclusion, our study revealed some novel insights into the gene expressions, cell-cell interactions, and intestinal microbiome profiles across different states of colitis, i.e., AC and CC, in comparison to HC. We identified previously unreported changes, particularly highlighting interactions between macrophages and epithelial cells/fibroblasts, mediated by chemokines and chemokine receptors. These findings were corroborated at the protein level, demonstrating altered expression of eNAMPT and subunits of the NOX2-complex in macrophages. It offers a clearer understanding of alternately polarized macrophages, particularly within the context of CC. Moreover, comparing AC and CC with HC, we observed notable differences in the composition of the intestinal microbiome. Specifically, CC exhibited distinct metabolic pathways, particularly in nicotinate and nicotinamide metabolism that may lead to colon cancer. Our findings shed new light on the pathophysiology of colitis and underscore the importance of considering both acute and chronic stages in understanding disease progression.

## Materials and methods

### Animal preparation
**Animals.** Wild-type C57BL/6 female mice (6 weeks old) were purchased from Koatech (Pyeongtaek, Korea). All animals were maintained in a specific pathogen-free (SPF) condition on the basis of the Center for Laboratory Animal Science, Hanyang University (Ansan, Korea). All experiments were approved by the Institutional Animal Care and Use Committee of Hanyang University (protocol 2020-0275) and were performed according to institutional guidelines provided by the Korean Ministry of Health and Welfare.

**DSS-induced colitis.** DSS-induced acute or chronic colitis mouse model were generated using 6-week-old C57BL/6 female mice (Koatech, Pyeongtaek, Korea), as previously described [https://doi.org/10.3390/antiox10121954]. To estimate the induction of acute colitis, mice were supplied with 3% (w/v) dextran sodium sulfate (molecular weight: 36,000–50,000 kDa, MP Biomedicals, Santa ana, CA, USA) dissolved in drinking water given ad libitum. The DSS solutions were produced freshly per 2 days. Control mice not fed DSS were administered sterile distilled water. For the chronic colitis model, mice were treated with 3% DSS for 6 days in 3 cycles and drinking water for 5 days between cycles, as illustrated in Fig. 1a. Compared to the animal's original body weight, the humane endpoint requiring euthanasia for weight loss is 20%. Weight loss cannot exceed 20% without an approved exception request.

**Tissue preparations.** Intestinal tissues were prepared from C57BL/6 female mice after induction of the acute or chronic colitis model and homogenized for scRNA-seq analysis.

### Sequencing
**DNA isolation and 16 S rRNA microbial analysis.** DNA was isolated using an Axen™ Total DNA mini-Kit (Axen™, Seoul, Korea), and DNA samples were then used in polymerase chain reaction (PCR). The 16 S universal primers 27, (forward) 5′-GAGTTTGATCMTGGCTCAG-3′; 518 (reverse) 5′-WTTACCGCGGCTGCTGG-3′ were used for the amplification of 16 S rRNA genes with a Axen™ H Taq PCR PreMix (Axen™, Seoul, Korea)[69]. After amplification, sequencing was performed on a NovaSeq X plus (Illumina, San Diego, CA, USA) by Macrogen (Seoul, Korea).

**Single-cell RNA sequencing and preprocessing.** We used Chromium Next GEM Single Cell 3p RNA library v3.1. After dissociation, single cells, reagents, and a single Gel Bead containing barcoded oligonucleotides were encapsulated into nanoliter-scale GEMs (Gel Bead in emulsion) using the Next GEM Technology. We then sampled a pool of about 3,500,000 10x Barcodes to separately index each cell's transcriptome and cell surface protein by partitioning thousands of cells into nanoliter-scale Gel Beads-in-emulsion (GEMs), where all generated DNA molecules share a common 10x Barcode.—The poly(dT) primer captures polyadenylated mRNA and barcoded, full-length cDNA is produced. Single Cell 3′ GEX and feature barcode libraries were sequenced on the Illumina sequencing system. For paired-end sequencing, two reads were sequenced from both ends of the fragment. Library preparation and sequencing were conducted by Macrogen (Seoul, Korea). CellRanger v6.1.1 was used to preprocess the sequencing reads with the genome provided by 10x genomics™ and obtain the standard cell-by-gene count matrix.

### Microbiome data processing
We processed the raw data of the microbiome following the standard procedure using DADA2. Additionally, we performed subsetting, aggregation, and filtering of the microbiome dataset using the phyloseq package. For alpha diversity analysis and manipulation, we utilized the miaRverse package, and visualization was conducted using miaViz. Data visualization through NMDS plots was carried out using the ANOSIM Test. Furthermore, we predicted the metagenome functions of the microbiome using the PICRUSt2 tool.

### Single-cell RNA-seq data processing
**Cell filtering.** With the gene expression count matrix, SCANPY[70] was used to filter cells. We discarded the cells of which the number of genes expressed were higher than 6000 or the percentage of mitochondrial gene (Hugo symbols starting with MT-) expression were higher than 15%.

**Preprocessing.** For down-stream analysis, we took some standard preprocessing steps. First, the count matrix was normalized to sum up to $10^4$ for each cell and then underwent log transformation using the *log1p* function. 2000 highly variable genes were then obtained using the *highly_variable_genes* function from SCANPY. Subsequently, dimension reduction using principal component analysis (PCA) with the number of PCA components of 15, discovery of neighboring cells with the number of neighbors of 10 and Leiden clustering with the default resolution were performed in that order.

**Cell type annotation.** With the preprocessed cell-by-gene count matrix, cell-types were annotated using HiCAT[71], a marker-based cell-type annotation tool, with the default parameters. The cell-type markers to run HiCAT were obtained from the R&D systems (https://www.rndsystems.com/resources/cell-markers). The list of subset markers can be found in Supplementary Data S2. We annotated 7 major types (T cells, B cells, myeloid cells, fibroblast, endothelial cells, epithelial cells, and smooth muscle cells), 12 minor types (CD4 + /CD8 + T cells, ILC, NK cells, B cells, Plasma cells, macrophages, DCs, fibroblast, endothelial cells, epithelial cells, and smooth muscle cells) and 33 subsets, including 7 CD4 + T cell subsets (Tfh, Th1, Th2, Th17, Th9, Th22, and Treg), 5 macrophage subsets (M1, M2a, M2b, M2c and M2d), 7 ILC subsets (ILC1, ILC2, NCR + ILC3, NCR- ILC3, ILCreg, and LTi), and 4 B cell subsets (memory, marginal zone, follicular and regulatory B cells). The minor types were used for cell-cell interaction inference and the subset results were used to explore the condition-specific subset composition of CD4 + T cells and macrophages. Fibroblast, endothelial cells, epithelial cells, and smooth muscle cells were not further divided into minor types and subsets as our primary focus was on immune cells.

**Inference of cell-cell interaction.** Cell-to-cell interaction analysis was performed using CellPhoneDB[27] (version 4.0.0), separately for each sample. Since the CellPhoneDB version 4.0.0 does not contain eNampt-NOX2 complex interaction, we added it to the DB, following the instructions at https://github.com/ventolab/CellphoneDB/tree/master/notebooks. The eNampt(Visfatin)-NOX2 complex interaction was

noticed by Xia et. al.[72] and can be found in the NOD-like receptor signaling pathway from the Kyoto Encyclopedia of Genes and Genomes https://www.genome.jp/kegg/. We used minor type for cell-cell interaction analysis since, with subsets, the number of cells were too small for some of the cell types. We selected ligand-receptor interactions with their *p*-value smaller than or equal to 0.05 and collected only those commonly found in 3 or more samples for each condition. Further, we collected ligand-receptor interactions commonly found in all three conditions and removed them from those for each condition in order to identify condition-specific ligand-receptor interactions in Fig. 2. All the cell-cell interactions commonly found in each condition were listed in Supplementary Data S1.

**DEG and GSE analysis**. DEG analysis were performed for each minor-types and subsets using SCANPY[70] ('rank_genes_groups' method), where we compared for each pair of conditions, i.e., AC versus HC, CC versus HC, and CC versus AC. To avoid sample-bias, we limited the number of cells from a sample no greater than twice the smallest number from a sample of the same type. Using *t*-test, we selected DEGs with their *p*-value smaller than or equal to 0.01 to be used for the gene set enrichment analysis (GSEA). Rather than to use adjusted *p*-value, we used non-adjusted one to obtain as many DEGs as possible. The prerank method in GSEApy[73] was used with the DEGs found with SCANPY and the Kyoto Encyclopedia of Genes and Genomes (KEGG) database for mouse. The pathways with their *p*-value smaller than or equal to 0.05 were chosen to explore the biological implication in the acute and chronic colitis.

## Data availability

The raw single-cell RNA-seq data was deposited to the gene expression omnibus (https://www.ncbi.nlm.nih.gov/geo/) and can be accessed by the accession number, GSE264408. The processed single-cell RNA-seq data in AnnData format was deposited to figshare.com and can be downloaded at https://figshare.com/articles/dataset/single-cell_RNA-seq_data_with_annotations_mouse_with_ulcerative_colitis_85_000_cells_/24670038. The raw and processed microbiome data was attached as supplementary data (Supplementary Data S3, S4, and S5). The single-cell RNA-seq data was processed using our single-cell data analysis pipeline, SCODA (https://mlbi-lab.net).

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

## Acknowledgements

This work was supported by Basic Science Research Program through the National Research Foundation of Korea (NRF) funded by the Ministry of Education, Science and Technology (NRF-2021R1A4A5032463). We thank Prof. Young-Ho Ahn, Dept. of Molecular Medicine, Ewha woman's university hospital, for helping us in dissociation of mice colon tissue for single-cell RNA-seq.

## Author contributions

S.Y. and C.S.Y. designed the overall experiments and devised the key idea. H.K.K. and C.Y. prepared mice and tissue samples. W.Y., M.K., and D.H. performed single-cell and microbiome data analysis. S.Y. guided data analysis. D.H. and C.S.Y. gave interpretation and feedback on the results. All authors wrote, read, and approved the final manuscript.

## Competing interests

The authors declare no competing interests.
