## [Peer Review File · Communications Biology]

Reviewers' comments:

Reviewer #1 (Remarks to the Author):

The paper by Hong D. et al. entitled "Integrative analysis of ulcerative colitis progression using single-cell RNA-seq and microbiome". UC is a comorbidity that affects the large intestine, especially the colon. The authors use an outstanding approach to identify concurrent changes in the composition of different cell populations and its relationship to changes in intestinal physiology and microbiota in different, i.e., acute, chronic, experimental settings of UC. Both hypothesis and objectives look well aligned.

Major comments:

-Methods: the procedure to determine "colitis score" was not mentioned in the methods section. If the space dedicated to methods details is restricted, detailed methods could be provided in a supplementary file.

-Results: the authors referred to rectal bleeding (lines 109); this is not clearly shown if "colitis score" refers to rectal bleeding (Figure 1A or 1C).

-Results: intestinal tract is often surrounded by adipose tissue; was this aspect considered during mRNA purification and thus HiCAT analysis when comparing the material from mice with different degrees of adiposity?

-Results: the determination of a selection of circulating biomarkers of intestinal barrier status (i.e., zonulin or others) should be also measured to confirm the relative gene expression changes in CC vs AC of Figure 1E.

-Results: line 131. As such the "restoration" of intestinal epithelial cell function is not supported by data; the authors just showed a relative partial increase in the proportion of epithelial cells in CC vs AC; another point is that the SD in the right panel of figure 1F?

-Results: line 148; and discussion: lines 329-on. The use of the term "communication", as synonym of crosstalk to describe the extent of association among different cell types should be softened; indeed, this was built on the basis that a collection of target molecules that were predominantly expressed by a certain cell type. Therefore the use of this terms should be changed by associations, or something to that effect. The authors should be cautious.

-Results: lines 171-172. "... , although statistically insignificant" means that there were not differences between groups.

-Results: which is the exact nature of the interaction between the extracellular form of NAMPT (eNAMPT) and CYBB and TLR4 substrates? Actually, eNAMPT is an enzyme that is produced also by other tissues, including the adipose tissue. Some studies suggest that it could behave as an adipokine. Were

the levels of eNAMPT determined? The authors did not demonstrate an increase of eNAMPT, but a relative increase in the synthesis of NAMPT.

-Results: was NAD⁺ or nicotinamide intermediates determined? Beyond its role as cofactor in redox metabolism, NAD⁺ is also a key coenzyme of the NAD⁺-dependent protein deacetylases (collectively defined as sirtuins). One member of this class of enzymes, SIRT1, can directly influence NFκB signaling and exert anti-inflammatory actions in target tissues. Actually, on the relative proportion of deacetylated SOD or other SIRT-1 substrates in intestinal tissue. The authors may attribute chronic inflammation to excess NAD⁺ synthesis in this setting, but it is not supported by data.

-Results and discussion: a pilot study showing concurrent changes in the composition of gut microbiota in subjects with inflammatory bowel diseases, like Crohn's, could add some translationality to this study. In this regard, the use of the mouse model and experimental design to assess chronic UC is adequate as it resembles the human disease. Whether the composition of gut microbiota under chronic UC has been previously reported in affected subjects? The authors did not deepen into this aspect in the discussion section.

Minor comments:

- Use the abbreviations consistently throughout the manuscript.

- Figure legends (general comment): Annotation is deficient in all figure legends: the meaning of acronyms used, e.g., the meaning of the abbreviation DW in figure 1C is not defined, DW was deduced to be distilled water as I read in lines 392-393 that "Control mice not fed DSS were administered with distilled water". The legends in all figures should provide enough information to warrant full reader's comprehension. The number of independent replicates should be shown. Likewise, the legends of supplementary figures are poorly described (Figure S1) or lacking (Figures S2, S3 and S4). Panels in each supplementary figure should be adequately identified.

- Figure 1, panel A, the x-axis of the right panel showing data on colitis score is not shown.

- Figure 4; panels C and D, some names are overlapped and therefore critical information are not adequately shown. The numbers in the x-axis are not aligned with relative data in columns of each panel. The y-axis is not shown.

- Figure 4; panel E, the family name of dark red bacteria is not shown.

- Check English as several errors and mistypings were detected. I show some examples as follows:

Line 102: the punctuation sign should be removed

line 109-111: the sentence "The symptoms of colitis, such as ... significantly suppressed in DSS-induced mice compared with control mice (Figure 1A and 1C)" please check if the verb "suppressed" is coherent.

Lines 198-200: revise the meaning of this sentence.

line 261: to compared  to compare

lines 268-269: "We also observations closely reflected tha data from patients...". Please check the meaning of this sentence.

line 345: UC compared HC  UC compared to HC

Reviewer #2 (Remarks to the Author):

Using scRNA-seq and microbial sequencing, this paper describes a complex cellular microenvironment and intestinal microbiome during UC. Overall, the cell-cell interaction during UC and the relationship between immune cells and microbiome analysis is interesting. However, the significance of their conclusions are largely impaired by the lack of necessary experiments to validate the key findings in this paper.

Major concerns:

1. Almost all the findings found in this paper are descriptive. The authors should at least experimentally validate 1-2 most important conclusions drawn from this paper. More importantly, authors should carefully compare their findings with previously published papers and clarify their add of values based on previously known facts on UC.
2. The authors claimed they used Smart-seq2 to profile single cell transcriptomes. However, given that the throughput of Smart-seq2 for scRNA-seq is very low, it is beyond imagination to sequence more than 80,000 cells using this technique. In the Method section, the authors described the 10X Chromium scRNA-seq approach instead of Smart-seq2, which is also very confusing.

Minor concerns:

1. The full name of DSS should be given in the first place it was mentioned in the Introduction.
2. Days after DSS treatment is missing in Figure 1A.
3. "rectal bleeding" is suppressed in DSS model is illogical in line 109.

Reviewer #1 expertise: intestinal inflammation

Reviewer #1 (Remarks to the Author):

The paper by Hong D. et al. entitled "Integrative analysis of ulcerative colitis progression using single-cell RNA-seq and microbiome". UC is a comorbidity that affect the large intestine, especially the colon. The authors use an outstanding approach to identify concurrent changes in the composition of different cell populations and its relationship to changes in intestinal physiology and microbiota in different, i.e., acute, chronic, experimental settings of UC. Both hypothesis and objectives look well aligned.

We thank the reviewer for such a detailed comments and positive suggestions. Most of all, as suggested by the reviewers and the editor, we have performed some small experiments to validate our arguments, which were more or less unclear with only the gene expression data from single-cell RNA-seq experiments. They became much clearer than before with the protein level measurements. During the revision, we also found many sentences were unclearly described and sometime incomplete, causing misunderstanding of our original intention. Please check our revision and line-by-line response below. We believe the manuscript is now more readable and complete for the reader of the journal.

Major comments:

-Methods: the procedure to determine "colitis score" was not mentioned in the methods section. If the space dedicated to methods details is restricted, detailed methods could be provided in a supplementary file.

-Results: the authors referred to rectal bleeding (lines 109); this is not clearly shown if "colitis score" refers to rectal bleeding (Figure 1A or 1C).

We have added how to assess colitis score in the supplemental file as follows. The scoring criteria take rectal bleeding into account.

Assessing colitis score

For clinical score of colitis, Body weight, occult or gross blood lost per rectum, and stool consistency were determined every other day during the colitis induction based on the scoring system shown in Supplemental Table 1 (Clin Microbiol Rev. 2002;15:79–94; Dig Dis Sci. 1993;38:1722-34). Weight loss was defined as the difference between initial and final weights, and diarrhea as the absence of fecal pellet formation and the presence of continuous fluid fecal material in the colon. Rectal bleeding was assessed based on the presence of diarrhea containing visible blood and on the presence of gross rectal bleeding. Clinical score of colitis values were calculated as ((weight loss score) + (diarrhea score) + (rectal bleeding score))/4. The clinical score was assessed by three trained investigators blinded to the treatment groups who were not aware of the treatment.

Supplemental Table 1 Criteria for Disease Activity Index

Score	Weight loss (%)	Stool Consistency	Bloodstain or gross Bleeding
0	None	Normal	Negative
1	1-5	Loose stool	Negative
2	5-10	Loose stool	Positive
3	10-15	Diarrhea	Positive
4	>15	Diarrhea	Gross bleeding

-Results: intestinal tract is often surrounded by adipose tissue; was this aspect considered during mRNA purification and thus HiCAT analysis when comparing the material from mice with different degree of adiposity?

In this work, we did not consider adipose tissue because (1) adipocytes are very hard to dissociate for single-cell sequencing, and (2) adiposity is known to be associated primarily with Crohn's disease but not strongly with ulcerative colitis. We are afraid that it would be quite difficult to consider adipocytes and their impact on UC in our revision since we sequenced only lamina propria, where adipocytes are hardly found.

-Results: the determination of a selection of circulating biomarkers of intestinal barrier status (i.e., zonulin or others) should be also measured to confirm the relative gene expression changes in CC vs AC of Figure 1E.

-Results: line 131. As such the "restoration" of intestinal epithelial cell function is not supported by data; the authors just showed a relative partial increase in the proportion of epithelial cells in CC vs AC; another point is that the SD in the right panel of figure 1F?

As the reviewer noted, we have performed a small experiment to measure the levels of proteins related to intestinal barrier status, including Zonula-1, Claudin-1, and Occludin. We have added the results in Figure 1H and a note on it as follows:

"We confirmed that the protein levels of intestinal barrier status biomarkers, including Zonula-1, Claudin-1 (known as cldn), and Occludin (known as ocln) shows the disruption of epithelial barrier in both AC and CC mice, while it was slightly restored in CC compared to AC mice (Figure 1H)"

We also replaced the barplot in Figure 1F with a boxplot, which shows quartiles, instead of standard deviation. A note was added in the text as follows.

"Although the difference in the proportion of epithelial cells were not statistically significant due to the limited number of samples, the epithelial barrier status could also be measured in terms of gene and protein level expressions of some related genes."

-Results: line 148; and discussion: lines 329-on. The use of the term "communication", as synonym of crosstalk to describe the extent of association among different cell types should be softened; indeed, this was built on the basis that a collection of target molecules that were predominantly expressed by a certain cell type. Therefore, the use of this terms should be

changed by associations, or something to that effect. The authors should be cautious.

In Result line 148, we have changed communications to ‘inferred cell-cell interactions’ to avoid any misunderstanding. Also, in discussion, we changed communication to interaction and added a note on cell-cell interactions at the end of 4th paragraph in the discussion as follows.

“Note that the inferred cell-cell interactions are based solely on the expression level of a gene or a set of genes, not the true measure of interactions. Even with this limitation, however, the inferred interaction gives us a valuable insight, at least, into overall changes in gene expressions and its possible implication in interactions between different cell types.”

-Results: lines 171-172. "... , although statistically insignificant" means that there were not differences between groups.

It is true that when we say statistical significance, it requires, in general, that p-value less than or equal to 0.05. Here, we wanted to say that, even though we cannot say it is statistically significant due to limited number of samples, we can see that macrophages are more or less ‘tend’ to be polarized into M2b in AC. To make it clearer, we have revised the sentence as follows:

“In AC, macrophages exhibited differentiation into M2B phenotype (Figure 3B). Although the population differences were not statistically significant due to the limited number of samples, the tendency shows the increased population of M2b in AC, especially when compared to CC. (Figure 3B). Different from AC, on the other hand, CC was characterized by a pronounced polarization towards the pro-inflammatory macrophage M1 phenotype, indicative of heightened inflammation (Figure 3A, 3B, and S2B).”

-Results: which is the exact nature of the interaction between the extracellular form of NAMPT (eNAMPT) and CYBB and TLR4 substrates? Actually, eNAMPT is an enzyme that is produced also by other tissues, including the adipose tissue. Some studies suggests that it could behave as an adipokine. Were the levels of eNAMPT determined? The authors did not demonstrate an increase of eNAMPT, but a relative increase in the synthesis of NAMPT.

In our previous study [15], we have conjectured that increased interaction between Visfatin (eNAMPT) and NOX2 complex (consisting of CYBB, CYBA, NCF1/2/4) might be a cause of chronic activation of NLRP3 inflammasome keeping the level of IL1b consistently high.

Since the finding was already published in [15], we just briefly touched it here. Probably the very brief and unclear description might cause the confusion. To clarify the point, we have added the detailed points around Cybb, eNampt and Tlr4 in the additional supplemental files as follows.

“Chronic activation of NLRP3 inflammasome

Our main points in [13] were (Supplemental Figure S3A)

- 1) Increased interaction between Visfatin (eNAMPT) and NOX2 complex (consisting of CYBB, CYBA, NCF1/2/4) increase the ROS level in macrophages*
- 2) Increased ROS level activate NLRP3 inflammasome to cleaves pro-IL1 β to mature IL1 β . (Activation signaling)*
- 3) eNAMPT-Tlr4 interaction induce NF- κ B-mediated NLRP3 and pro-IL-1 β expression. (Priming signaling)*

To check if they really happen in DSS-induced UC mice, specifically in chronic UC, we first checked the expression of NAMPT and of genes comprising the NOX2-complex, including

Cybb, Cyba, Ncf1, 2, and 4 in macrophages (Supplemental Figure S3B). As expected from our previous findings, they were mostly over-expressed significantly, except Nampt and Ncf1. However, even with the insignificance of Nampt and Ncf1, the tendencies look similar to those of other genes. Increased ROS (reactive oxygen species) level can be implicitly confirmed from the over-expression of ROS related genes, such as Sod2, Neat1, and Hif1a (Supplemental Figure S3C). The genes involved in priming of NLRP3 inflammasome, such as Tlr4, Nfkb1, and Nlrp3 were also significantly over-expressed in CC compared to HC (Supplemental Figure S3C). Of note, the increased expression level does not necessarily mean only the increased production rate in individual macrophage, but also the increase in the fraction of macrophages expressing these genes, i.e., alternately polarized macrophages. The violin plots in Supplemental Figure S3E clearly shows the higher fraction of macrophages are expressing those genes around NLRP3 inflammasome in chronic colitis than those in healthy colon.

Certainly, the increase in the number of RNAs does not necessarily mean the increase in protein level, e.g., the over-expression of IL1 β gene in RNA level, which is actually pro-IL1 β , does not necessarily mean higher level of IL-1 β secretion. Therefore, we measured protein level expression of those genes from both colon lysates (Supplemental Figure S3F) and colon macrophages (Supplemental Figure S3G) separately. In both cases, the tendencies we found from RNA level could be confirmed in protein level too.”

Supplemental Figure S3. Expression profiling related to the activation of the NLRP3 inflammasome. **A:** Schematic diagram of NAMPT's binding with TLR4 and NOX2 complex-mediated NLRP3 inflammasome activation. **B:** Violin plots for the expression of NAMPT, CYBB, and TXNIP in macrophages. **C:** Violin plots for the expression of ROS-related genes in macrophages. **D:** Western blotting result showing the protein level expression of indicated markers. **E:** Violin plots for the expression of genes associated with NLRP3 inflammasome activation

-Results: was NAD⁺ or nicotinamide intermediates determined? Beyond its role as cofactor in redox metabolism, NAD⁺ is also a key coenzyme of the NAD⁺-dependent protein deacetylases (collectively defined as sirtuins). One member of this class of enzymes, SIRT1, can directly influence NFκB signaling and exert anti-inflammatory actions in target tissues. Actually, the relative proportion of deacetylated SOD or other SIRT-1 substrates in intestinal tissue. The authors may attribute chronic inflammation to excess NAD⁺ synthesis in this setting, but it is

not supported by data.

This comment seems to lie on the same line as the previous comments. We could identify the over-expression of ROS-scavenging and hypoxia-alleviating genes, including SOD2, NEAT1, and HIF1A, but not SIRT1 (Figure S3C). We summarized their implication in a new subsection “The overexpression of eNAMPT and NOX2 subunits leads to the progression of severe colitis induced by DSS.”

-Results and discussion: a pilot study showing concurrent changes in the composition of gut microbiota in subjects with inflammatory bowel diseases, like Crohn's, could add some translationality to this study. In this regard, the use of the mouse model and experimental design to assess chronic UC is adequate as it resembles the human disease. Whether the composition of gut microbiota under chronic UC has been previously reported in affected subjects? The authors did not deepen into this aspect in the discussion section.

As suggested by the reviewer, we added a translational analysis of microbiota metagenome data between AC and HC, as well as CC and HC in the middle of “The intestinal microbiome undergoes changes as colitis progresses” subsection as follows:

“We next compared metagenome between AC and HC, as well as CC and HC (Figure 4D). Although the metagenomes of AC and HC showed similar distributions of AC microbiota (represented by the orange color), indicating little difference in diversity, there were variations in the types of microbiota belonging to the family, as shown on the left side of Figure 4D. Conversely, the coverage range of the CC metagenome decreased relative to that of HC, indicating a reduced microbiota diversity in CC, as shown on the right side of Figure 4D. Healthy individuals have a majority of Lactobacillaceae (approximately 54.6%) in their microbiota, a family within the phylum Firmicutes (Figure 4E). The *Lactobacillus* genus is a prominent member of this family, demonstrates protective effects in ulcerative colitis^{40, 41}. The AC metagenome exhibited a significant decrease in Lactobacillaceae but a marked increase in Turicibacteraceae compared to the HC (Figure 4E). A significant decrease in Lactobacillaceae, which is also known to alleviate damage to the intestinal epithelial cell barrier, likely contributes to the disruption of the epithelial barrier in AC (Figure 1 and 4E)⁴⁰. Analysis of the intestinal microbiome composition indicated a progressive elevation in Turicibacteraceae in AC (17.9%) compared to healthy states (0.7%), with no such increase observed in CC (2.9%) (Figure 4E and 4F). Imbalance of microbiota such as Lactobacillaceae and Turicibacteraceae may predispose individuals to acute colitis compared to HC. Furthermore, the abundance of Bacteroidaceae and Enterobacteriaceae increased only in CC (39.3% and 10.7% respectively) (Figure 4E and 4F). Numerous proposed roles of Bacteroidaceae in colitis pathogenesis have been identified. Animal studies demonstrate the crucial role of *Bacteroides* in exacerbating colitis symptoms⁴²⁻⁴⁴. Additionally, extensive

research highlights the significant contribution of Enterobacteriaceae to IBD pathogenesis⁴⁵. It seems that a reduction in microbiota diversity and alterations in the abundance of Bacteroidaceae and Enterobacteriaceae may accelerate chronic colitis, eventually resulting in sustained dysbiosis and an altered pathogenic microbiota ratio, which may ultimately lead to colon cancer.”

Minor comments:

- Use the abbreviations consistently throughout the manuscript.

As noted by the reviewer, we included the abbreviations after acknowledgments and before the references.

- Figure legends (general comment): Annotation is deficient in all figure legends: the meaning of acronyms used, e.g., the meaning of the abbreviation DW in figure 1C is not defined, DW was deduced to be distilled water as I read in lines 392-393 that "Control mice not fed DSS were administered with distilled water". The legends in all figures should provide enough information to warrant full reader's comprehension. The number of independent replicates should be shown. Likewise, the legends of supplementary figures are poorly described (Figure S1) or lacking (Figures S2, S3 and S4). Panels in each supplementary figure should be adequately identified.

As noted by the reviewer, we included the abbreviations for DW (distilled water) and DSS (Dextran sodium sulfate) in Figure 1 legend. Also, we have revised all the figure legend providing detailed information on the results and added figure legends for Figures S2, S3 and S4, which were omitted unintentionally before.

- Figure 1, panel A, the x-axis of the right panel showing data on colitis score is not shown.

As mentioned by the reviewer, we have added an x-axis label to the right panel to indicate "Days after 3% DSS TX".

- Figure 4; panels C and D, some names are overlapped and therefore critical information are not adequately shown. The numbers in the x-axis are not aligned with relative data in columns of each panel. The y-axis is not shown.

We resolved the overlapping names in Figure 4C and 4D (now they are 5C and 5D) by using legends.

- Figure 4; panel E, the family name of dark red bacteria is not shown.

It was unidentified bacteria. We specified the family name of dark red as "unknown".

- Check English as several errors and mistypings were detected. I show some examples as follows:

We have thoroughly reviewed the text for errors and corrected any mistakes or typos that were found.

Line 102: the punctuation sign should be removed

We have removed the punctuation mark.

line 109-111: the sentence "The symptoms of colitis, such as ... significantly suppressed in DSS-induced mice compared with control mice (Figure 1A and 1C)" please check if the verb "suppressed" is coherent.

We have revised the sentence as follows:

"The symptoms of colitis, such as weight loss, rectal bleeding, and edema¹², were observed in DSS-induced mice compared with control mice."

Lines 198-200: revise the meaning of this sentence.

To clarify the meaning of the sentence, we have updated the description of CYBB (NOX2 subunit), eNAMPT, and NLRP3 activation in both Figure 4 and Supplementary Figure S3, where we have also provided relevant references for the updates. Additionally, we have included some new figures on the expression levels of NOX2 complex subunits and NAMPT obtained through scRNA-seq analysis. Furthermore, we have verified them using western blotting. Please see Figure 4, which was newly added.

line 261: to compared  to compare

Thank you for the pinpoint. We corrected the typo

lines 268-269: "We also observations closely reflected the data from patients...". Please check the meaning of this sentence.

We have revised the paragraph as follows:

"We observed an increase in the numbers of Treg, Th17, and Tfh cells within inflamed tissues of patients with UC, mirroring the profile observed in CC. However, non-inflamed tissue did not show significant differences in cell counts (Figure 6C and 6D)."

(During the revision of our manuscript, we added new results, and therefore, we moved the original Figure 5C and D to Figure 6C and D.)

line 345: UC compared HC  UC compared to HC

corrected as suggested

Reviewer #2 expertise: single-cell analysis, multi-omics

Reviewer #2 (Remarks to the Author):

Using scRNA-seq and microbial sequencing, this paper describes a complex cellular microenvironment and intestinal microbiome during UC. Overall, the cell-cell interaction during UC and the relationship between immune cells and microbiome analysis is interesting. However, the significance of their conclusions are largely impaired by the lack of necessary experiments to validate the key findings in this paper.

We thank the reviewer for the comments and suggestion to perform necessary experiments. As suggested by the reviewer and the editor, we have performed several experiments to validate our arguments, which were more or less unclear with only the gene expression data from single-cell RNA-seq experiments. They become much clearer than before with the protein level measurements. Please check our revision and response below. We believe the manuscript is now more readable and complete for the reader of the journal.

Major concerns:

1. Almost all the findings found in this paper are descriptive. The authors should at least experimentally validate 1-2 most important conclusions drawn from this paper. More importantly, authors should carefully compare their findings with previously published papers and clarify their add of values based on previously known facts on UC.

According to the reviewer's comments, we performed some small experiments on the following issues.

- 1) Measure protein level expression of Zonula-1, Claudin-1 and Occludin to check epithelial barrier status of HC, AC and CC
- 2) Measured protein level expression of proteins including secreted IL1b, internal and external NAMPT and proteins comprising NOX2-complex, i.e., Cybb, Cyba, Ncf1, Ncf2 and Ncf4. To check that they are mostly from macrophages, we sorted macrophages only and measure their protein level expression again.
- 3) Also measured protein level expression of Sod2, Sirt1 and NFkB to compare ROS state in each condition.

Regarding the second comments, to emphasize our contribution compared to the previously known facts on UC, we added concluding remarks at the end of discussion section as follows.

"In conclusion, our study reveals novel insights into the gene expressions, cell-cell interactions, and intestinal microbiome profiles across different states of colitis, i.e., AC and CC, in comparison to HC. We identified previously unreported changes, particularly highlighting interactions between macrophages and epithelial cells/fibroblasts, mediated by chemokines and chemokine receptors, including NAMPT and CYBB. These findings were corroborated at the protein level, demonstrating altered expression of eNAMPT and subunits of the NOX2-complex in macrophages. It offers a clearer understanding of alternately polarized macrophages, particularly within the context of CC. Moreover, comparing AC and CC with HC, we observed notable differences in the composition of the intestinal microbiome. While both AC and CC showed similar diversity indices, CC exhibited distinct metabolic pathways, particularly in nicotinate and nicotinamide metabolism. Our findings shed new light on the pathophysiology of colitis and underscore the importance of considering both acute and chronic stages in understanding disease progression."

2. The authors claimed they used Smart-seq2 to profile single cell transcriptomes. However, given that the throughput of Smart-seq2 for scRNA-seq is very low, it is beyond imagination to sequence more than 80,000 cells using this technique. In the Method section, the authors described the 10X Chromium scRNA-seq approach instead of Smart-seq2, which is also very confusing.

We certainly used 10X genomics' Chromium technology to perform single-cell RNA-seq experiments. We have corrected.

Minor concerns:

1. The full name of DSS should be given in the first place it was mentioned in the Introduction.

We have added the full name of DSS in the first place in the introduction.

2. Days after DSS treatment is missing in Figure 1A.

We have added the ticks in x-axis of the right figure in panel A (days).

3. "rectal bleeding" is suppressed in DSS model is illogical in line 109.

It was a typo. We have revised the sentence as follows:

"The symptoms of colitis, such as weight loss, rectal bleeding, and edema¹², were observed in DSS-induced mice compared with control mice."

REVIEWERS' COMMENTS:

Reviewer #1 (Remarks to the Author):

The authors have addressed all the queries.

Reviewer #2 (Remarks to the Author):

The Authors have addressed all of my concerns with the original manuscript. The revised manuscript is ready for publication.